# Feature-Level Adversarial Attacks and Ranking Disruption for Visible-Infrared Person Re-identification

**Xi Yang[1], Huanling liu[1], De Cheng[1]\*, Nannan Wang[1], Xinbo Gao[2]**
[1]Xidian University, [2]Chongqing University of Posts and Telecommunications
yangx@xidian.edu.cn,huanlingliu@stu.xidian.edu.cn, dcheng@xidian.edu.cn
nnwang@xidian.edu.cn, gaoxb@cqupt.edu.cn

## Abstract

Visible-infrared person re-identification (VIReID) is widely used in fields such as video surveillance and intelligent transportation, imposing higher demands on model security. In practice, the adversarial attacks based on VIReID aim to disrupt output ranking and quantify the security risks of models. Although numerous studies have been emerged on adversarial attacks and defenses in fields such as face recognition, person re-identification, and pedestrian detection, there is currently a lack of research on the security of VIReID systems. To this end, we propose to explore the vulnerabilities of VIReID systems and prevent potential serious losses due to insecurity. Compared to research on single-modality ReID, adversarial feature alignment and modality differences need to be particularly emphasized. Thus, we advocate for feature-level adversarial attacks to disrupt the output rankings of VIReID systems. To obtain adversarial features, we introduce *Universal Adversarial Perturbations* (UAP) to simulate common disturbances in real-world environments. Additionally, we employ a *Frequency-Spatial Attention Module* (FSAM), integrating frequency information extraction and spatial focusing mechanisms, and further emphasize important regional features from different domains on the shared features. This ensures that adversarial features maintain consistency within the feature space. Finally, we employ an *Auxiliary Quadruple Adversarial Loss* to amplify the differences between modalities, thereby improving the distinction and recognition of features between visible and infrared images, which cause the system to output incorrect rankings. Extensive experiments on two VIReID benchmarks (i.e., SYSU-MM01, RegDB) and different systems validate the effectiveness of our method.

## 1 Introduction

VIReID is widely applied in key tasks such as security monitoring[1–3]. Suppose the law enforcement agency of a city uses ReID system to monitor public places for tracking criminal suspects. Internal personnel may attempt to deceive the system by modifying the images [4–6]of criminal suspects due to improper behavior or other reasons, in order to protect specific individuals. According to Figure1, infrared adversarial samples will erroneously match visible samples, while visible adversarial samples will also erroneously match infrared samples. The credibility and stability of VIReID are crucial in such special application scenarios. However, there is currently insufficient theoretical research on the security of VIReID. Therefore, this paper explores how to obtain better adversarial features and how to address the task characteristics of VIReID modality differences and re-ranking.

---

\*Corresponding author.

38th Conference on Neural Information Processing Systems (NeurIPS 2024).

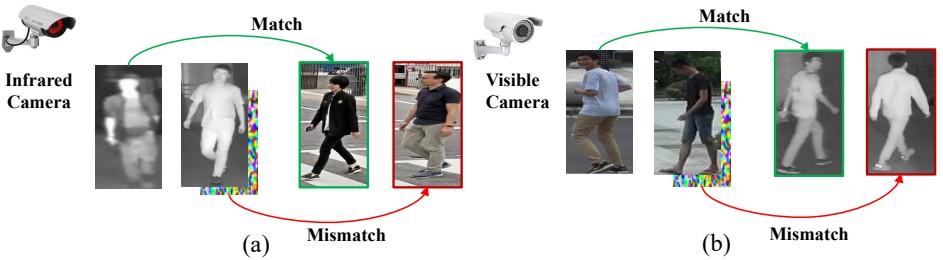

Figure 1: Security risks of VIReID in the physical world. Images with added noise are referred to as adversarial samples. Red indicates that adversarial samples will incorrectly match pedestrians. Green indicates that clean samples will correctly match pedestrians.

The current research on adversarial attacks in the fields of face recognition and person re-identification mainly focuses on digital attacks [7–9], which refer to the manipulation, distortion, or tampering of digital images to deceive, disrupt, or mislead systems, leading to erroneous identification results or reduced recognition accuracy[10]. Adversarial attacks based on ReID focus on crafting adversarial samples suitable for visible images and disrupting internal rankings. In contrast, VIReID requires more consideration regarding the generalizability of attack methods under different imaging mechanisms and how to align adversarial features between infrared and visible images. Moreover, dealing with two modalities necessitates comparing distances between modalities and within modalities. To address these challenges, we propose a feature-level adversarial attacks and ranking disruption for VIReID.

First, we adopt the method of universal adversarial perturbation (UAP)[11] to generate adversarial samples, which seeks a set of universal perturbations independent of the image and can generalize well in deep neural networks. At the same time, it significantly lowers the threshold for implementing adversarial attacks and adapts to different systems. Secondly, to make visible and infrared images more consistent in the feature space, we propose a frequency-spatial attention module, achieving adversarial feature alignment by unifying frequency and spatial features. Visible and infrared images are generated under different imaging conditions; the former provides rich texture information, while the latter contains significant pixel amplitude information. This module uses fast Fourier transform to decompose features into amplitude and phase, corresponding to the texture details and spatial position information of images, respectively. Since these features are closely related to the spatial domain, a spatial attention module is chosen for these two components to further emphasize or suppress different regions in the feature map. Additionally, a weighted spatial attention module is applied to the shared features to maintain consistency in the feature space. This method not only focuses on the frequency domain features of visible and infrared images from different imaging conditions but also emphasizes their spatial features for significant pedestrian poses, thereby achieving adversarial feature alignment. To disrupt the ranking of identification results, we propose an auxiliary quadruple adversarial loss function. The visible and infrared pedestrian features extracted by the first-stage model are used as auxiliary features in the calculation process of the features loss function extracted in the second stage. By pulling the distance between the same modality and the same person closer and pushing the distance between different modalities and the same person farther, while also ensuring that the intra-class distance is smaller than the inter-class distance, the differences between modalities are expanded and the ranking under different modalities is disrupted. By utilizing generated features containing multiple types of information, the network's ability to explore features at different levels is enhanced. That is, with four types of features, we ensure a double guarantee to achieve the goal of disrupting the ranking. The main contributions of this paper can be summarized as follows:

- We are the first to propose exploring the security of VIReID, considering the alignment of adversarial features across modalities in VIReID.

- We propose a frequency-spatial attention module that integrates frequency-domain features with spatial features to enhance the consistency and representation ability of adversarial features.

- We design an auxiliary quadruple adversarial loss function, which utilizes auxiliary features to amplify the differences within and between modalities, thereby disrupting the ranking results.

## 2  Related Work

**Visible-Infrared Person Re-identification.** VIReID [12–15]refers to the technique of identifying and matching pedestrians from one modality to another using visible or infrared images. Moreover, VIReID finds wide application in the field of security, enhancing the intelligence level of security systems. The significant differences between different modalities make VIReID challenging. To alleviate the modality discrepancy at the feature level, some methods adopt single-stream, dual-stream, or multi-stream networks[16–19], extracting shared features from different modalities by designing various attention mechanisms and loss functions[20–23]. Ye et al.[16] proposed the concept of metric learning, jointly optimizing modality-specific and modality-shared matrices. Subsequently, Zhu et al.[18] proposed a hetero-center loss, which for the first time shortens the distance between feature centers of the same identity, bridging the gap between features of the same pedestrian across different modalities. In addition, Ling et al.[24] devised a multi-constraint similarity learning approach to comprehensively explore the relationships between cross-modal information. Meanwhile, as the posture and shape of pedestrians provide important information in the recognition process, these methods mainly focus on spatially enhancing feature representation. However, there are also differences in frequency information between visible and infrared images. Li et al.[25] proposed a novel frequency-domain modality-invariant feature learning framework to reduce modality differences from a frequency-domain perspective.

**Adversarial Attacks.** In the fields of computer graphics and pattern recognition, adversarial attacks[26–29] are an important area of research. Many studies have revealed the vulnerability of deep models to carefully crafted small perturbation, resulting in significant errors with high confidence in predictions. Adversarial attacks aim to explore adversarial noise that causes deep learning models to behave abnormally. FGSM[30] belongs to the single-step attack algorithm, which optimizes the loss by quickly determining the direction of perturbation for input samples and calculates the adversarial perturbation through backpropagation. The concept of iterative thinking was subsequently incorporated into FGSM, thereby leading to the development of Projected Gradient Descent [31] (PGD). Subsequently, adversarial attacks on ReID were also studied, with the core idea of generating well-crafted adversarial examples or disturbing ranking results. LTA[32] used local grayscale iteration to generate adversarial examples, mainly focusing on disturbing the color of the original image. A mis-ranking formula was proposed by DMR[33] to increase the distance between images of the same pedestrian while decreasing the distance between images of different pedestrians, effectively disrupting the ranking results.

**Cross-modality Attacks.** Adversarial instances are widely present across various domains of visible images. In recent years, exploration of adversarial instances in the field of infrared pedestrian detection has begun. Osahor et al.[34] discussed perturbation by altering pixel values within infrared images. Subsequently, Zhu et al.[35] attempted for the first time to alter the infrared radiation distribution of the human body by simulating additional heat sources using a set of small bulbs, generating physical adversarial examples. To be more easily implemented in the physical world, Wei et al.[36, 37] considered the different imaging mechanisms of visible and infrared sensors, proposing a unified adversarial patch to execute cross-modality physical attacks. Meanwhile, we chose to perform security evaluation on visible-infrared person re-identification models proposed in recent years in the digital world.

## 3  Proposed Method

As shown in Figure 2, the overall structure of the proposed method adopts a dual-stream ResNet network as the backbone. Firstly, *Universal Adversarial Perturbation* (UAP) are added separately to visible and infrared images. In the first stage, the *Frequency-Spatial Attention Module* (FSAM) is embedded to extract frequency-domain spatial correlated features as auxiliary features for images of two different modalities, visible and infrared. Subsequently, through a shared module, further focus is applied to the temporal domain features. This completes the focus on spatial features in both frequency and normal domains, making the learned information more diversified, thereby completing

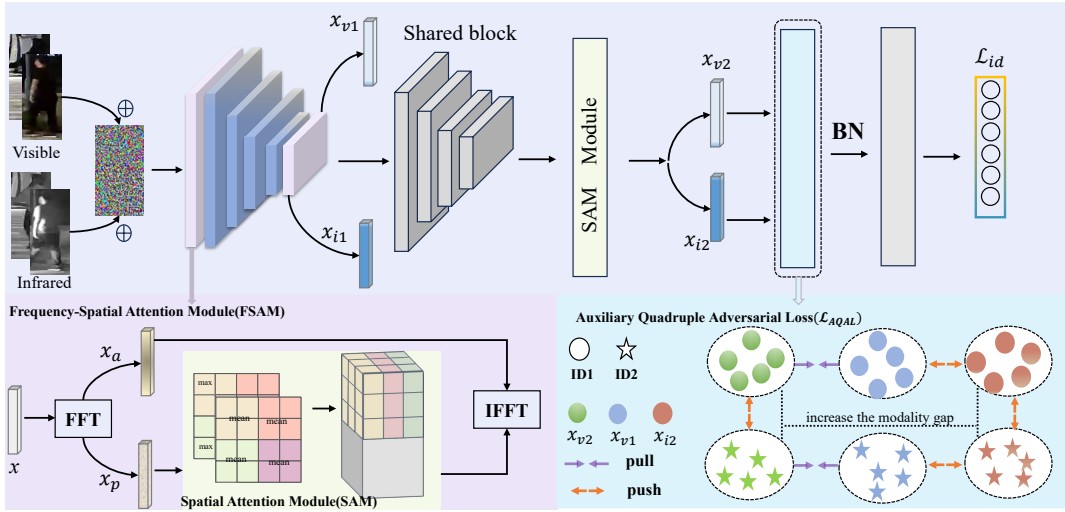

Figure 2: Overview of the proposed method. FSAM consists of FFT and IFFT along with a spatial focusing module, focusing on the frequency-domain spatial characteristics of the image. We propose a auxiliary quadruple adversarial loss function and provide a simple illustration of its operation.

the learning of features in the second stage. During the training phase, the features from the first stage are used as auxiliary features and combined with the features from the second stage, all of which are inputted into the *Auxiliary Quadruple Adversarial Loss* for optimizing the entire module.

### 3.1 Universal Adversarial Perturbation

Universal Adversarial Perturbation (UAP) aims to generate a single perturbation that can be added to any image from the same distribution, resulting in mis-classification when added. Deep neural networks are highly susceptible to this type of UAP, yet they remain imperceptible to the human eye. Defining a set of images $I$ that satisfies distribution $\mu$, where $g(I)$ fits the output function, and after perturbation $\delta$, the labels are not equal. That is:

$$g(I + \delta) \neq g(I). \tag{1}$$

When $\delta$ is constrained by two conditions simultaneously, the optimization problem of finding universal perturbation can be described as follows:

$$s.t. ||\delta||_p \leq \varepsilon, \quad P_{I \sim \mu}(g(I + \delta) \neq g(I)) \geq 1 - \epsilon, \tag{2}$$

where $P(\cdot)$ represents probability, $|| \cdot ||_p$ denotes the p-norm, $\varepsilon$ indicates the magnitude of the perturbation to ensure that the adversarial perturbation is visually imperceptible, $\mu$ represents the data distribution, $\epsilon \in (0,1]$ denotes the success rate of deception to ensure the attack's success rate. The goal is to find an adversarial perturbation $\delta$ that can be added to all sample points and will result in misclassification of adversarial samples with a probability of 1 - $\epsilon$. The UAP algorithm does not require solving optimization problems or gradients of the model and is applied in scenarios where a large number of adversarial samples need to be quickly generated.

So we choose universal perturbation for addition, which can be used on both visible and infrared images and can also adapt to scenarios with a large number of pedestrian samples in real-world environments. The introduction of UAP can make the model more independent of specific data distributions and training sets, thereby reducing the model's dependence on specific data and making the system more flexible and adaptable to challenges in different environments and scenarios. Visible and infrared images generate their respective adversarial samples as follows:

$$\hat{Q}_{vi} = Q_{vi} + \delta_{vi}, \tag{3}$$

$$\hat{Q}_{ir} = Q_{ir} + \delta_{ir}. \tag{4}$$

For the input $Q_{vi}$ and $Q_{ir}$, adding perturbation($\delta_{vi}, \delta_{ir}$) related to the data distribution to each, the generated adversarial sample $\hat{Q}_{vi}, \hat{Q}_{ir}$ can deceive the system by exploiting visual similarity attacks.

## 3.2 Frequency-Spatial Attention Module

### 3.2.1 Fast Fourier Transform

The Fast Fourier Transform (FFT) is widely utilized in the field of image processing to convert images into the frequency domain, enabling the analysis of the frequency components of the image. This aids in understanding the overall structure, texture, and edge information of the image. Therefore, we combine the Fast Fourier Transform with a spatial attention module to focus on the unique features in the frequency domain of the image, which aids in enhancing feature representation capabilities. First, we provide a brief introduction to the basic concepts of the FFT. Given the feature $x \in \mathbb{R}^{C \times H \times W}$ output by network, its FFT can be expressed as follows:

$$\mathcal{F}(\boldsymbol{x})(u,v) = \frac{1}{\sqrt{HW}} \sum_{h=0}^{H-1} \sum_{w=0}^{W-1} x(h,w) e^{-2j\pi(u\frac{h}{H} + v\frac{w}{W})}, \tag{5}$$

where $j$ represents the imaginary unit, $u$ and $v$ are the horizontal and vertical coordinates of the $x$, and $\mathcal{F}(\cdot)$ denotes the Fourier transform, $C$, $H$ and $W$ denote the number of the channel, height and width of features. The frequency-domain feature $\mathcal{F}(\boldsymbol{x})$ is represented as $\mathcal{F}(\boldsymbol{x}) = \mathcal{R}(\boldsymbol{x}) + j\mathcal{I}(\boldsymbol{x})$, where $\mathcal{R}(\boldsymbol{x})$ and $\mathcal{I}(\boldsymbol{x})$ represent the real and imaginary part of $\mathcal{F}(\boldsymbol{x})$. These real and imaginary parts can be converted to amplitude and phase spectrums, which can be formulated as follows:

$$\mathcal{A}(\boldsymbol{x})(u,v) = \left[\mathcal{R}^2(\boldsymbol{x})(u,v) + \mathcal{I}^2(\boldsymbol{x})(u,v)\right]^{1/2}, \tag{6}$$

$$\mathcal{P}(\boldsymbol{x})(u,v) = \arctan\left[\frac{\mathcal{I}(\boldsymbol{x})(u,v)}{\mathcal{R}(\boldsymbol{x})(u,v)}\right]. \tag{7}$$

As shown in Figure 3, in the task of VI-ReID, the amplitude component captures the overall brightness and contrast of pedestrian images, reflecting the luminance and color information of the image, while the phase component captures the structural information and details of the pedestrians, including their shape and outline, to help distinguish between different pedestrians' details and features. The combination of these components allows for the effective extraction of global and local features of pedestrians. By focusing further on spatial information characteristics in the phase component, attention to spatial information in the frequency domain is increased, enhancing the ability to express distinguishing features. In the context of the features $x$ extracted by the network, we represent the amplitude component of the FFT as $\boldsymbol{x}_a$, and the phase component as $\boldsymbol{x}_p$. Given that the dual-branch network separately extracts visible and infrared pedestrian features,

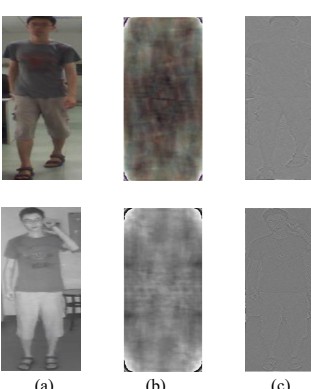

(a)      (b)      (c)

Figure 3: Decomposition and reconstruction of visible and infrared image in the frequency domain. (a) denote visible and infrared images of pedestrian; (b)present the reconstructed images with amplitude information only; (c) are the reconstructed images with phase information only.

the amplitude component of the visible pedestrian features after FFT is represented as $\boldsymbol{x}_{va}$ and the phase component as $\boldsymbol{x}_{vp}$, while the amplitude component of the infrared pedestrian features after FFT is represented as $\boldsymbol{x}_{ia}$ and the phase component as $\boldsymbol{x}_{ip}$.

### 3.2.2 Spatial Attention Module

The spatial attention module generates spatial attention maps using the spatial relationships within the features. It focuses on the distribution of information, locating the position and shape of the subject while reducing background information interference. We apply max pooling and average pooling operations along the channel axis. These two pooling operations respectively extract the maximum and average value information from the feature map and concatenate them to form a richer feature descriptor. Pooling is performed along the channel axis, which helps highlight information-rich

regions in the feature map and better locate key information. In summary, the specific calculation process is as follows:

$$M(\boldsymbol{x}) = Sigmoid(f([MaxPool(\boldsymbol{x}); AvgPool(\boldsymbol{x})])), \tag{8}$$

where $M(\boldsymbol{x})$ denotes the weight distribution generated by applying the convolution layer and *Sigmoid* denotes the sigmoid function. $f$ represents a convolution operation with the filter size of $7 \times 7$. $MaxPool(\boldsymbol{x})$ and $AvgPool(\boldsymbol{x})$ represent the features after passing through the max-pooling layer and average-pooling layer, respectively.

Given that the phase component involves more spatial information, we sequentially apply the spatial attention module to obtain the new phase component($\boldsymbol{x}_{vp}^{'}$,$\boldsymbol{x}_{ip}^{'}$), which can be expressed as follows:

$$\boldsymbol{x}_{vp}^{'} = M_v(\boldsymbol{x}_{vp}) \otimes \boldsymbol{x}_{vp}, \tag{9}$$

$$\boldsymbol{x}_{ip}^{'} = M_i(\boldsymbol{x}_{ip}) \otimes \boldsymbol{x}_{ip}, \tag{10}$$

where $\otimes$ denotes element-wise multiplication, $M_v(\boldsymbol{x}_{vp})$ and $M_i(\boldsymbol{x}_{ip})$ represent the attention weights generated by the feature phase components. The combination of the original amplitude component and the new phase component can reconstruct the original feature information through inverse fast Fourier transform(IFFT). At this point, the network outputs features $\boldsymbol{x}_{v1}$ and $\boldsymbol{x}_{i1}$, which will serve as auxiliary features. This process can be described as:

$$x_{v1} = IFFT(x_{vp}^{'}, x_{va}), \tag{11}$$

$$x_{i1} = IFFT(x_{ip}^{'}, x_{ia}). \tag{12}$$

The features are then further input into a shared module $\mathcal{T}$ in the model to complete feature extraction. At this point, choosing to pass through the spatial attention module allows for focusing on spatial information in the original domain. This two-stage process completes the focus on both frequency domain and original domain spatial information, enhancing the representation capability of distinguishing features. Thus, the final extracted visible and infrared features can be expressed as:

$$\boldsymbol{x}_{v2} = M_v^{'}(\mathcal{T}(\boldsymbol{x}_{v1})) \otimes \mathcal{T}(\boldsymbol{x}_{v1}), \tag{13}$$

$$\boldsymbol{x}_{i2} = M_i^{'}(\mathcal{T}(\boldsymbol{x}_{i1})) \otimes \mathcal{T}(\boldsymbol{x}_{i1}), \tag{14}$$

where $M_v^{'}$ and $M_i^{'}$ indicates the weight distribution generated by the second-stage SAM module, resulting in the final stage features, namely $\boldsymbol{x}_{v2}$ and $\boldsymbol{x}_{i2}$.

### 3.3 Auxiliary Quadruple Adversarial Loss

We propose an auxiliary quadruplet adversarial loss function to disrupt the system's output ranking. This method effectively adapts to ReID issues by attacking the predicted ranking results. Additionally, considering the modality differences involved in VIReID, we introduce auxiliary features to fully leverage the information disparities between modalities, erroneously outputting cross-modality ranking results. In this process, there are both modality differences and identity differences. Therefore, we approach it from two aspects: under the condition of the same modality, minimizing the distance between the same identities and maximizing the distance between different identities; under the condition of the same identity, minimizing the distance between the same modalities and maximizing the distance between different modalities. Let's start by controlling the same modality case:

$$\mathcal{L}(\boldsymbol{x}_{v2}, \boldsymbol{x}_{v1}) = \left[ D(\boldsymbol{x}_{v2}^j, \boldsymbol{x}_{v1}^j) - D(\boldsymbol{x}_{v2}^j, \boldsymbol{x}_{v2}^k) \right]_+, \tag{15}$$

where $\boldsymbol{x}_{v1}^j$ represents the auxiliary feature, $j$ and $k$ denote different pedestrians, $D(\cdot, \cdot)$ represents the Euclidean distance between two feature vectors and $[x]_+ = max(x, 0)$. When the modalities are the same, the task is transformed into a single-modality ReID. This problem discusses the disruption of matching pairs across modalities, while still maintaining the original requirement for the same modality, namely, encouraging that the maximum distance between the most easily identifiable pairs of images across identities is still less than the minimum distance between the most easily identifiable pairs of images within an identity, ensuring that the sorting output within the same modality is normal.

This part ensures that the original order and matching relationships within the modality are not disrupted.

When the identities are consistent across modalities:

$$\mathcal{L}(\boldsymbol{x}_{v2}, \boldsymbol{x}_{v1}, \boldsymbol{x}_{i2}) = \left[ D(\boldsymbol{x}_{v2}^j, \boldsymbol{x}_{v1}^j) - D(\boldsymbol{x}_{i2}^j, \boldsymbol{x}_{v1}^j) \right]_+ , \qquad (16)$$

we encourage reducing the distance between pedestrians in the same modality while increasing the distance between pedestrians in different modalities. This strategy aims to narrow the distance within the same modality while widening the distance between different modalities, thereby increasing the modalities differences. It disperses the features in the feature space, making it more challenging for the model to cluster features. This is intended to impact the matching and sorting results across modalities. The loss function augmented with visible features can be expressed as:

$$\mathcal{L}(\boldsymbol{x}_{v2}, \boldsymbol{x}_{i2}, \boldsymbol{x}_{v1}) = \sum_{\substack{j,k=1 \\ j \neq k}}^{N} \left[ D(\boldsymbol{x}_{v2}^j, \boldsymbol{x}_{v1}^j) - D(\boldsymbol{x}_{i2}^j, \boldsymbol{x}_{v1}^j) - D(\boldsymbol{x}_{v2}^j, \boldsymbol{x}_{v2}^k) + \alpha \right]_+ , \qquad (17)$$

where $N$ is the number of person ID in a mini-batch. Meanwhile, using more discriminative embedding centers($\boldsymbol{c}_{v1}, \boldsymbol{c}_{v2}, \boldsymbol{c}_{i1}, \boldsymbol{c}_{i2}$) for each class, we introduce a margin term $\alpha$ to balance the two terms. Thus, the loss function augmented with visible features and infrared features can be expressed as follows:

$$\mathcal{L}(\boldsymbol{c}_{v2}, \boldsymbol{c}_{i2}, \boldsymbol{c}_{v1}) = \sum_{\substack{j,k=1 \\ j \neq k}}^{N} \left[ D(\boldsymbol{c}_{v2}^j, \boldsymbol{c}_{v1}^j) - D(\boldsymbol{c}_{i2}^j, \boldsymbol{c}_{v1}^j) - D(\boldsymbol{c}_{v2}^j, \boldsymbol{c}_{v2}^k) + \alpha \right]_+ , \qquad (18)$$

$$\mathcal{L}(\boldsymbol{c}_{i2}, \boldsymbol{c}_{v2}, \boldsymbol{c}_{i1}) = \sum_{\substack{j,k=1 \\ j \neq k}}^{N} \left[ D(\boldsymbol{c}_{i2}^j, \boldsymbol{c}_{i1}^j) - D(\boldsymbol{c}_{v2}^j, \boldsymbol{c}_{i1}^j) - D(\boldsymbol{c}_{i2}^j, \boldsymbol{c}_{i2}^k) + \alpha \right]_+ . \qquad (19)$$

Finally, the auxiliary quadruplet adversarial loss function ($\mathcal{L}_{AQAL}$) is ultimately formulated as:

$$\mathcal{L}_{AQAL} = \mathcal{L}(\boldsymbol{c}_{v2}, \boldsymbol{c}_{i2}, \boldsymbol{c}_{v1}) + \mathcal{L}(\boldsymbol{c}_{i2}, \boldsymbol{c}_{v2}, \boldsymbol{c}_{i1}). \qquad (20)$$

The $\mathcal{L}_{AQAL}$ forces the distance between modalities to increase, preventing them from easily clustering together, thereby disrupting the overall ranking results.

### 3.4 Objective Function

Besides the auxiliary quadruple adversarial loss $\mathcal{L}_{AQAL}$, we also have the identity loss $\mathcal{L}_{id}$. The training process of VIReID is considered an image classification problem, where each identity is a distinct class. During the testing phase, the output from the pooling layer or embedding layer is used as the feature extractor. Given an input image $x_i$ with label $y_i$, the probability of $x_i$ being recognized as class $y_i$is encoded using the softmax function and denoted as $p(y_i|x_i)$. The identity loss is then computed by the cross-entropy

$$\mathcal{L}_{id} = -\frac{1}{N} \sum_{i=1}^{N} log(p(y_i|x_i)), \qquad (21)$$

where N represents the number of training samples within each batch.

## 4 Experiments

### 4.1 Implementation Details

The experiments are conducted on an NVIDIA GeForce 3090 GPU with Pytorch. We chose the powerful baseline model AGW[38], which is a ResNet-50 pretrained on ImageNet, as the backbone network. During training, we randomly sampled 16 identities, each with 4 images, to form a mini-batch of size 64. Pedestrian images are resized to 288 × 144. Data augmentation included random

Table 1: Comparison of CMC (%) and mAP (%) with the state-of-the-art methods on SYSU-MM01 and RegDB datasets. Our results show the best results in terms of Rank-1 accuracy and mAP .

| methods | SYSU-MM01 | | | | | | | | RegDB | | | | | |
| | All-search | | | | Indoor-search | | | | Visible to Thermal | | | Thermal to Visible | | |
| | Rank-1 | Rank-10 | mAP | mINP | Rank-1 | Rank-10 | mAP | mINP | Rank-1 | mAP | mINP | Rank-1 | mAP | mINP |
|---|---|---|---|---|---|---|---|---|---|---|---|---|---|---|
| Before Attack | 47.50 | 84.39 | 47.65 | 35.30 | 54.17 | 91.14 | 62.97 | 59.23 | 70.05 | 66.37 | - | 70.49 | 65.90 | - |
| FGSM [30] | 36.02 | 59.22 | 31.80 | 19.72 | 33.25 | 69.17 | 46.31 | 30.36 | 45.87 | 44.39 | 36.59 | 46.15 | 46.12 | 43.24 |
| PGD [31] | 26.60 | 46.65 | 25.67 | 16.90 | 36.89 | 75.19 | 43.50 | 27.20 | 29.37 | 26.87 | 17.83 | 29.05 | 29.12 | 17.14 |
| SMA [39] | 21.72 | 39.80 | 21.34 | 20.39 | 25.77 | 51.24 | 30.91 | 29.96 | 19.85 | 17.37 | 9.49 | 16.57 | 18.23 | 10.84 |
| UAP [11] | 17.59 | 56.81 | 25.35 | 20.89 | 35.34 | 47.86 | 30.75 | 19.13 | 29.51 | 24.42 | 16.64 | 19.61 | 18.95 | 12.67 |
| LTA [32] | 15.47 | 30.39 | 17.71 | 13.44 | 21.68 | 38.56 | 26.61 | 20.59 | 11.60 | 10.86 | 6.07 | 14.56 | 13.11 | 7.75 |
| DMR [33] | 9.20 | 24.43 | 10.21 | 4.71 | 13.62 | 27.14 | 14.94 | 6.27 | 4.97 | 4.80 | 2.12 | 6.09 | 5.26 | 3.54 |
| Ours | **0.79** | **9.83** | **2.81** | **1.69** | **1.68** | **17.35** | **6.73** | **5.65** | **0.49** | **0.85** | **0.57** | **0.71** | **0.89** | **0.60** |

Table 2: Comparison of CMC (%) and mAP (%) of different VIReID systems before and after attack. Bold numbers indicate values after attack.

| methods | SYSU-MM01 | | | | | | RegDB | | | | | |
| | All-search | | | Indoor-search | | | Visible to Thermal | | | Thermal to Visible | | |
| | Rank-1 | Rank-10 | mAP | Rank-1 | Rank-10 | mAP | Rank-1 | Rank-10 | mAP | Rank-1 | Rank-10 | mAP |
|---|---|---|---|---|---|---|---|---|---|---|---|---|
| HCLoss[18] | 56.96/**1.09** | 91.50/**10.37** | 54.95/**2.91** | 59.74/**2.09** | 92.07/**18.85** | 64.91/**7.17** | 86.02/**2.67** | 96.36/**10.73** | 74.80/**2.82** | 87.28/**1.02** | 97.04/**4.66** | 78.30/**2.07** |
| CAJ[40] | 69.88/**1.04** | 95.71/**10.32** | 66.89/**2.99** | 76.26/**2.45** | 97.88/**18.93** | 80.37/**7.66** | 85.0/**1.04** | 95.5/**10.32** | 84.6/**2.99** | 88.3/**2.45** | 98.5/**18.93** | 81.9/**7.66** |
| MMN [41] | 70.60/**1.39** | 96.2/**4.55** | 66.9/**3.81** | 76.2/**4.26** | 97.2/**27.81** | 79.6/**8.16** | 91.6/**0.49** | 97.7/**0.97** | 84.1/**1.71** | 87.5/**3.50** | 96.0/**14.08** | 80.5/**3.72** |
| DEEN [42] | 74.70/**1.71** | 97.60/**10.49** | 71.80/**3.55** | 80.30/**2.04** | 99.00/**17.96** | 83.30/**7.34** | 91.1/**3.15** | 97.8/**12.48** | 85.1/**4.05** | 89.5/**2.85** | 96.8/**16.34** | 83.4/**6.21** |

horizontal flipping and random erasing with a probability of 0.5. We optimize using the stochastic gradient descent (SGD) optimizer, with a weight decay set to 0.0005 and a momentum parameter set to 0.9. The initial learning rate for both datasets was set to 0.1, and it was decayed by a factor of 0.1 at the 20th and 50th epochs, respectively. Finally, the margin $\alpha$ in the auxiliary quadruplet adversarial loss function is set to 0.2. We adopt a warm-up learning rate scheme, with a total of 60 training epochs.

## 4.2 Datasets

SYSU-MM01[1] is a cross-modality pedestrian re-identification dataset proposed in 2017. There are 287,628 visible images and 15,792 infrared images in total. Cameras 1 and 2 are installed in well-lit environments, cameras 3 and 6 operate under infrared conditions, and cameras 4 and 5 are placed in outdoor scenes. The dataset comprises two testing modes: all-search and indoor-search. The all-search mode is more challenging because the gallery includes images from all cameras.

The RegDB [43]dataset comprises 412 individuals, with each person having 20 images, including 10 visible images and 10 infrared images. Among the 412 individuals, there are 254 females and 158 males, with 156 individuals captured in frontal views and 256 individuals captured in rear views. During the testing phase, RegDB offers two modes: visible to infrared and infrared to visible. In the visible to infrared mode, visible images serve as query images, while infrared images are used as gallery images. The infrared to visible mode operates in the opposite manner.

## 4.3 Comparison with State-of-the-Art Methods

We select six different methods to evaluate the security of the AGW model. Among them, FGSM and PGD are typical gradient-based methods for generating adversarial samples, while UAP generates universal perturbations that are independent of the image. Additionally, we chose SMA, DMR, and LTA, three pedestrian re-identification attack methods which primarily disrupt ranking and generate adversarial samples through local color iteration. The results, as shown in Table1, indicate a sharp decline in all evaluation metrics after attacking AGW across two datasets and two different tests. In the RegDB dataset, under two testing modes, our mAP decreased dramatically from 66.37% and 65.90% to 0.85% and 0.89%, nearly approaching 0. This demonstrates that our method is more suitable for testing robustness against VIReID tasks.

To demonstrate the universality of our method, we select three approaches to improve the recognition accuracy of VIReID systems: HCLoss enhances intra-class cross-modality similarity through a heterogeneous center loss function, CAJ improves recognition accuracy based on data augmentation techniques, and MMN and DEEN utilize different network designs to better extract effective shared features. These methods are commonly used to improve VIReID accuracy at the feature level. As

Table 3: Analysis about the influence of each component in terms of Rank-1 (%) and mAP (%).

| Noise | FSAM | SAM | $\mathcal{L}_{AQAL}$ | SYSU-MM01 | | RegDB | |
|---|---|---|---|---|---|---|---|
| | | | | Rank-1 | mAP | Rank-1 | mAP |
| | | | | 46.73 | 45.78 | 82.24 | 76.52 |
| ✓ | | | | 16.67 | 18.28 | 33.00 | 30.60 |
| ✓ | ✓ | | | 12.75 | 12.17 | 15.74 | 12.91 |
| ✓ | | ✓ | | 11.04 | 13.19 | 20.73 | 19.81 |
| ✓ | | | ✓ | 14.62 | 15.52 | 15.86 | 15.53 |
| ✓ | ✓ | ✓ | | 5.10 | 6.35 | 7.58 | 7.51 |
| ✓ | ✓ | | ✓ | 1.07 | 3.27 | 0.58 | 1.35 |
| ✓ | | ✓ | ✓ | 1.74 | 3.81 | 6.70 | 5.77 |
| ✓ | ✓ | ✓ | ✓ | **0.79** | **2.81** | **0.49** | **0.85** |

Table 4: Performance comparison of different feature extraction methods in terms of CMC (%) and mAP (%) on RegDB. (Setting: Baseline + SFM + $\mathcal{L}_{AQAL}$.)

| Settings + FSAM | SYSU-MM01 | | | | RegDB | | | |
|---|---|---|---|---|---|---|---|---|
| | Rank-1 | Rank-10 | mAP | mINP | Rank-1 | Rank-10 | mAP | mINP |
| block0 | 1.05 | 10.60 | 3.15 | 2.04 | 0.49 | 4.17 | 1.07 | 1.27 |
| block1 | 1.05 | 10.52 | 4.29 | 5.33 | 0.53 | 1.69 | 1.68 | 2.41 |
| block2 | 1.05 | 10.07 | 3.56 | 2.25 | 0.54 | 2.94 | 1.23 | 1.11 |
| block3 | 1.34 | 10.41 | 4.07 | 4.72 | 0.49 | **1.46** | 1.67 | 2.41 |
| block4 | 1.01 | 10.12 | 2.91 | **1.68** | 0.49 | 4.71 | 0.93 | 0.67 |
| block0-block4(ours) | **0.79** | 9.83 | **2.81** | 1.69 | **0.48** | 4.64 | **0.85** | **0.57** |

shown in Table2, our experimental results show that after applying our attack method, the mAP of all systems dramatically decreased across different datasets and test modes (for example, the Rank-1 of DEEN drops from 74.70% to 1.71%, and mAP falls from 71.80% to 3.55% in all-search mode), indicating that these systems cannot withstand our attacks. This demonstrates that our method can effectively test the robustness of various VIReID systems.

## 4.4 Ablation Study

As shown in Table3, the effectiveness of different modules is validated on two datasets. Using the SYSU-MM01 dataset as an example, after adding noise to generate adversarial samples, the mAP decreases from 45.75% to 18.28%. Subsequently, the three modules are individually tested on this basis, and all show a decrease in accuracy, with the FSAM module being the most effective, reducing the mAP to 12.17%. Additionally, since both FSAM and SAM are designed to enhance adversarial feature representation capabilities, the combined effect of these two modules is tested, further reducing the mAP to 6.35%. The experiments demonstrate that all three proposed modules are highly effective. Although targeting different aspects, their combined usage enhances the effectiveness of method.

Determining which stage of ResNet-50 should have the FSAM module inserted. In this experiment, we use ResNet-50 as the backbone, which has five stages: block 0 to block 4. We study the impact on model performance by inserting the FSAM module after different stages of ResNet-50. This experiment controls only the insertion position of FSAM as the variable, keeping all other factors constant. As shown in Table4, inserting FSAM after block 0 and block 4 results in lower accuracy, indicating better performance. This suggests that in the shallow layers of the network, image processing captures more effective information from the initial frequency domain features, while in the deeper layers of the network, the feature representation capability is further enhanced, and spatial information becomes more important. Based on these results, we integrate the proposed FSAM module after block 0 and block 4 of the ResNet-50 model.

As shown in Table 5, we compared our frequency domain attention module with channel attention mechanisms like SE-Net [44] and ECA-Net[45]. The results reveal that while SE-Net and ECA-Net excel in certain areas, our frequency domain attention module is highly competitive across various metrics. It achieves the best Rank-1 and mAP scores on both datasets. Channel attention mechanisms often focus on global features, which may overlook local details and spatial relationships in cross-modal VIReID tasks, leading to potential information loss and reduced performance.

Table 5: Performance of the FSAM module replacing different attention mechanisms on CMC(%) and mAP(%).

| Attention | SYSU-MM01 | | | | RegDB | | | |
|---|---|---|---|---|---|---|---|---|
| | Rank-1 | Rank-10 | mAP | mINP | Rank-1 | Rank-10 | mAP | mINP |
| SENet | 1.66 | 12.15 | 3.30 | 1.76 | 0.53 | 5.00 | 1.15 | 0.64 |
| ECA-Net | 1.60 | 11.65 | 3.24 | **1.65** | 0.68 | **3.64** | 1.20 | 0.68 |
| Ours) | **0.79** | **9.83** | **2.81** | 1.69 | **0.48** | 4.64 | **0.85** | **0.57** |

## 4.5 Visualization Analysis

We compare the t-SNE visualization results on the baseline and the proposed method. To ensure fairness, we randomly select several images of ten identities from all cameras. For each individual, 20 visible images and 20 infrared images are randomly chosen. As shown in Figure4, after the attack, the clustering of all pedestrians became more dispersed, with the visible and infrared modalities each forming their own clusters, and the relative distance between them increasing. The visualization experiments validated the effectiveness of our attack method, as the attack further exaggerated the gap between the two modalities, effectively suppressing the output performance. These results demonstrate that our method is highly effective for security testing in VIReID.

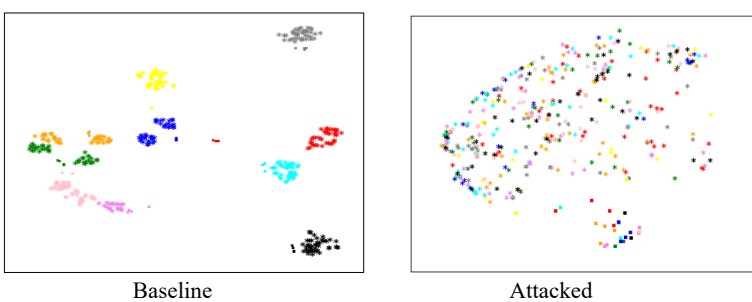

Baseline          Attacked

Figure 4: t-SNE visualization comparison before and after the attack. Different colors represent different identities. The 'asterisks' and 'rectangles' denote the infrared person features and visible person features, respectively. After the attack, the features are dispersed, enlarging the distance between modalities.

## 5 Conclusion

In this paper, our objective is to conduct security validation of VIReID systems and propose a novel attack method suitable for cross-modality tasks. We introduce the frequency-spatial attention module, which are used at two stages of feature extraction, focusing on spatial information in both frequency and source domains to enhance the representation capability of adversarial features and strengthen the effectiveness of adversarial samples. Additionally, we propose an auxiliary quadruple adversarial loss function considering the modalities differences involved in VIReID tasks to interfere with the ranking of system outputs, completing the robustness test of the current VIReID system. Extensive experiments not only deepens the understanding of the security of cross-modality ReID systems but also provides a new direction for the development of VIReID and emphasizes the importance of ensuring their reliability and protection in practical applications.

## 6 Acknowledgement

This work was supported in part by the National Natural Science Foundation of China under Grant 62372348, Grant 62441601, Grant 62176195, Grant 62176198, Grant U22A2096, Grant U21A20514; in part by the Key Research and Development Program of Shaanxi under Grant 2024GX-ZDCYL-02-10; in part by Shaanxi Outstanding Youth Science Fund Project under Grant 2023-JC-JQ-53; in part by the Shaanxi Province Core Technology Research and Development Project under Grant 2024QY2-GJHX-11; in part by the Fundamental Research Funds for the Central Universities under Grant QTZX24080 and Grant QTZX23042.

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
