# OpenReview forum: "Feature-Level Adversarial Attacks and Ranking Disruption for Visible-Infrared Person Re-identification"
_NeurIPS.cc/2024/Conference — NeurIPS 2024 poster_

### Official Review · Reviewer_sAfC · 2024-07-07

**Soundness:** 4
**Presentation:** 3
**Contribution:** 3
**Rating:** 7
**Confidence:** 5

**Summary:**

There is currently a lack of research focused on the security of VIReID systems. In light of this, the authors are the first to propose a method to disrupt the output ranking of VIReID systems by leveraging feature-level adversarial attacks, considering the specific characteristics of VIReID. This paper introduces Universal Adversarial Perturbations (UAP) and adopts a Frequency-Spatial Attention Module (FSAM) to integrate frequency and spatial information, ensuring the consistency of adversarial features. Additionally, the authors propose an Auxiliary Quadruple Adversarial loss to amplify modality differences, thereby disrupting the system's output ranking results.

**Strengths:**

1.This study is the first to propose research on the security of VIReID systems, filling a gap in this field.
2.The motivation behind this paper is strong. By introducing UAP, the method is able to generate universal adversarial samples adaptable to different modalities. It employs the FSAM module to enhance the consistency of adversarial features between modalities. L_AQA is able to effectively disrupt the system's output ranking by amplifying modality differences. This method aligns with the characteristics of digital attacks and caters to the task requirements of VIReID.
3.The authors conducted extensive experiments on different datasets and with different backbone models, validating the effectiveness and generalizability of the method. The results indicate that the proposed method achieves state-of-the-art performance.
4.The paper is well-structured, logically clear, and well-organized.

**Weaknesses:**

1.The paper points out the differences from single-modality ReID research but does not clearly explain the specific distinctions between them. Additionally, it does not clarify why adversarial feature alignment and modality differences need to be emphasized.
2. What is the total loss function? What is the meaning of ‘L_id’ in Figure 2?
3.The authors suggest that the phase components of the features after Fourier transformation are more closely related to spatial information, but there are no corresponding visualization experiments to support this claim.

**Questions:**

1. In Section 3.2.2, what are the meanings of features x_v1 and x_i1, and how are they generated?
2. Regarding Table 2, the methods selected for attacking different VIReID systems are somewhat limited, which does not seem sufficient to validate the generalizability of the proposed approach. It is recommended to include more VIReID methods.
3. According to Table 3, the authors explore the effectiveness of different modules, but there are no experimental investigations into the effectiveness of the two specific modules, which I think is insufficient.

**Limitations:**

The authors explain the limitations of their work, and there is no negative societal impact.

---

> ### Author Rebuttal · Authors · 2024-08-07
>
> **w1 :Comparison with ReID**
>
> As illustrated in Figure 2, the differences and shared characteristics between visible and infrared pedestrian images can be understood. Compared to ReID, VIReID can extract features from two different types of images. While VIReID leverages the complementary information from both modalities, it also introduces additional modality-specific differences, such as the loss of crucial color information. However, the spatial information in both modalities remains unaffected and can mutually aid in capturing pedestrian features. By utilizing the FSAM, VIReID integrates global features from the frequency domain with detailed information from the spatial domain, thereby enhancing feature consistency and improving the representation capability of adversarial features.
>
> **w2: What is the total loss function? What is the meaning of ‘$\cal{L_{id}}$’ in Figure 2?**
>
> The total loss function can be formulated as follows:
>
> $$
> \cal{L_{total}} = \cal{L_{AQAL}} + \cal{L_{id}}
> $$
>
> $\cal{L_{id}}$ represents the identity loss. The training process of VIReID is considered an image classification problem, where each identity is a distinct class. During the testing phase, the output from the pooling layer or embedding layer is used as the feature extractor. Given an input image $x{i}$ with label $y{i} $, the probability of $x{i}$ being recognized as class $y{i}$is encoded using the softmax function and denoted as $p(y{i}|x_{i})$. The identity loss is then computed by the cross-entropy
>
> $$
> \begin{equation}
> 	{\cal{L_{id}}} = -\frac{1}{N}\sum_{i=1}^{N}log(p(y_{i}|x_{i})).
> \end{equation}
> $$
>
> where N represents the number of training samples within each batch.
>
> **w3: Visual experiments supporting FSAM**
>
> As shown in Figure 3, in the task of VI-ReID, the amplitude component captures the overall brightness and contrast of pedestrian images, reflecting the luminance and color information of the image, while the phase component captures the structural information and details of the pedestrians, including their shape and outline, to help distinguish between different details and features of pedestrians. The combination of these components allows for the effective extraction of global and local features of pedestrians. By focusing further on spatial information characteristics in the phase component, attention to spatial information in the frequency domain is increased, enhancing the ability to express distinguishing features.
>
> **Q1: the meanings of features $x_{v1}$ and $x_{i1}$**
>
> After applying the FSAM module to the spatial components of the original features, $x_{v1}$  and  $x_{i1}$ are combined with the amplitude components of the original features and then transformed back to the original domain using the inverse Fourier transform (IFFT). This process can be described as:
> $$
> \begin{equation}
> 	x_{v1} = IFFT(x_{vp}^{'},x_{va});
> \end{equation}
> $$
>
> $$
> \begin{equation}
> 	x_{i1} = IFFT(x_{ip}^{'},x_{ia});
> \end{equation}
> $$
>
> **Q2: Attacking more VIReID systems**
>
> We understand the reviewer's concern regarding the breadth of attack method validation. In this study, we chose the DDAG[1] and MRCN[2] systems as targets for the attacks. These two methods are representative in the existing VIReID field, employing dual attention mechanisms and modal restoration and compensation mechanisms, respectively, to reduce modality differences, covering different technical approaches. According to Table 4, although their modules provided some resistance, they still achieved very good results. This further demonstrates the broad applicability and effectiveness of our method.
>
> | Methods | SYSU-MM01      |                 |                | RegDB          |                |                |
> | ------- | -------------- | --------------- | -------------- | -------------- | -------------- | -------------- |
> |         | Rank-1         | Rank-10         | mAP            | Rank-1         | Rank-10        | mAP            |
> | DDAG    | 54.75/**1.65** | 90.39/**10.86** | 53.02/**3.27** | 69.34/**0.94** | 91.49/**9.70** | 63.46/**1.35** |
> | MRCN    | 70.80/**2.36** | 96.50/**11.18** | 67.30/**5.87** | 95.10/**2.36** | 98.80/**8.87** | 89.20/**1.16** |
>
> [1] Dynamic dual-attentive aggregation learning for visible-infrared person re-identification. ECCV 2020
>
> [2] MRCN: A Novel Modality Restitution and Compensation Network for Visible-Infrared Person Re-identification. AAAI 2023
>
> **Q3: Verify the effectiveness of different modules and their combination**
>
> We investigate the effectiveness of different modules and supplement our experiments with the effectiveness of each pair of specific modules. Comparing rows 5-7 with rows 1-3 in Table 5, our proposed modules are effective when used individually and show additive effectiveness when combined. Our experiments validate the superiority of the proposed method.
>
> | Noise | FSAM | SFM  | ${\cal{L}}_{AQAL}$ | SYSU-MM01 |          | RegDB    |          |
> | ----- | ---- | ---- | -------------------- | --------- | -------- | -------- | -------- |
> |       |      |      |                      | Rank-1    | mAP      | Rank-1   | mAP      |
> |       |      |      |                      | 46.73     | 45.78    | 82.24    | 76.52    |
> | ✔     |      |      |                      | 16.67     | 18.28    | 33.00    | 30.60    |
> | ✔     | ✔    |      |                      | 12.75     | 12.17    | 15.74    | 12.91    |
> | ✔     |      | ✔    |                      | 11.04     | 13.19    | 20.73    | 19.81    |
> | ✔     |      |      | ✔                    | 14.62     | 15.52    | 15.86    | 15.53    |
> | ✔     | ✔    | ✔    |                      | 5.10      | 6.35     | 7.58     | 7.51     |
> | ✔     | ✔    |      | ✔                    | 1.07      | 3.27     | 0.58     | 1.35     |
> | ✔     |      | ✔    | ✔                    | 1.74      | 3.81     | 6.70     | 5.77     |
> | ✔     | ✔    | ✔    | ✔                    | **0.79**  | **2.81** | **0.49** | **0.85** |

---

### Official Review · Reviewer_ksiy · 2024-07-10

**Soundness:** 2
**Presentation:** 2
**Contribution:** 2
**Rating:** 6
**Confidence:** 5

**Summary:**

There is currently a lack of research on the security of VIReID systems. This paper proposes to explore the vulnerabilities of VIReID systems and prevent potential serious losses due to insecurity. To obtain adversarial features, this paper introduces Universal Adversarial Perturbations (UAP) to simulate common disturbances in real-world environments. Additionally, the authors employ a Frequency-Spatial Attention Module (FSAM), integrating frequency information extraction and spatial focusing mechanisms, and further emphasize important regional features from different domains on the shared features. Extensive experiments on two VIReID benchmarks (i.e., SYSU-MM01, RegDB) and different systems validate the effectiveness of the proposed method.

**Strengths:**

Experimental results on the VI-Person ReID task show the proposed method works well.

**Weaknesses:**

1. Adversarial attack in VI-ReID is a meaningful research topic. However, merely utilizing the noise-added image as an adversarial sample is singular and impractical. How could a real-world application employ this method for an attack in a visible-infrared surveillance system? Numerous other methods for generating adversarial samples are not discussed in the paper, thereby significantly diminishing its value.
2. This paper does not present the image with the added noise. If the image loses its original information after noise is introduced, the recognition results will be poor even without using the method of this paper.
3. FSAM is not a new thing, which has been proposed in many works [1] and is not innovative.
[1] Li Y, Zhang T, Zhang Y. Frequency Domain Modality-invariant Feature Learning for Visible-infrared Person Re-Identification[J]. arXiv preprint arXiv:2401.01839, 2024.

**Questions:**

How could a real-world application employ this method for an attack in a visible-infrared surveillance system? The motivation for the methodology of this paper requires further elaboration

**Limitations:**

Authors need to consider more methods of attack than just adding noise.

---

> ### Author Rebuttal · Authors · 2024-08-07
>
> **w1 & limitation: Discussion of the definition of adversarial attacks and what they mean, and how they can be applied in the real world**
>
> Adversarial attacks induce misclassification in classifiers by introducing subtle perturbations to inputs. In 2014, Goodfellow et al. [1] demonstrated this using a panda image. Since then, research has focused on identifying model vulnerabilities and improving model robustness. Studies reveal that deep neural networks are susceptible to adversarial examples, where minor input perturbations cause incorrect predictions with high confidence. As deep learning applications expand, the security issues they expose garner more attention.
>
> Technologies like face recognition and person re-identification (ReID) have significant potential in security fields such as criminal investigation, person tracking, and behavior analysis. These technologies support the safe and stable operation of public society. However, their security and reliability are questioned in adversarial environments, limiting their specialized applications. Adversarial attacks offer a new perspective on system security, showing how adversarial examples can evade recognition or impersonate others. For human observers, these examples are often indistinguishable from legitimate ones but cause deep models to err. Evaluating recognition systems' robustness with adversarial attacks identifies system vulnerabilities, encouraging improvements in machine learning model robustness.
>
> Similarly, visible-infrared person re-identification (VIReID) is increasingly used in security systems, necessitating an exploration of its security. Existing attack methods focus on visible image features, neglecting other modalities and cross-modal data distribution variations, potentially reducing their effectiveness in cross-modal image retrieval. This study examines VIReID model security and proposes a universal perturbation attack designed for VIReID.
>
> [1] Explaining and harnessing adversarial examples. Arxiv  2014
>
> [2] Efficient Decision-based Black-box Adversarial Attacks on Face Recognition. CVPR 2019
>
> [3] Transferable, Controllable, and Inconspicuous Adversarial Attacks on Person Re-identification With Deep Mis-Ranking. CVPR 2020
>
> **w2: Visualization of antagonistic samples**
>
> As shown in the figure1, we can observe the before-and-after comparison of the visualization with the addition of UAP (Universal Adversarial Perturbation). Adversarial attacks involve applying perturbations that are imperceptible to the human eye, causing the model to produce incorrect outputs. The visualization results show no visible difference between adversarial samples and original samples. According to Table 5, although the images appear identical to the human eye after adding noise, they lose some of the original information, leading to decreased recognition performance.
>
> However, our goal is to simulate real-world interference, hoping to enhance the representation capability of adversarial features to handle more complex and diverse scenarios. This approach aims to provide a stronger evaluation capability when assessing the security of VIReID systems, thereby enhancing the social value of our research.
>
> **w3: FSAM is not a new thing**
>
> While both our method and FDMNet explore feature extraction in the frequency domain, they differ in objectives and model design. FDMNet aims for modal-invariant feature learning via frequency domain decomposition, using the Instance Adaptive Amplitude Filtering (IAF) and Phase Preservation Normalization (PPNorm) modules to enhance modal-invariant components, as we as suppressing modality-specific ones. In contrast, our FSAM module integrates frequency and spatial features for adversarial feature alignment, using Universal Adversarial Perturbations (UAP) to generate adversarial samples. FSAM unifies features by combining the frequency and spatial domains, making visible and infrared image features more consistent.
>
> In summary, FDMNet emphasizes the differences in amplitude components within the frequency domain, striving to enhance the consistency of modal features. Conversely, FSAM focuses on the commonality of phase components in the frequency domain, facilitating the alignment of adversarial features in both frequency and spatial domains.
>
> **Q:How could a real-world application employ this method for an attack in a visible-infrared surveillance system? The motivation for the methodology of this paper requires further elaboration.**
>
> This study presents a digital attack on VIReID, offering a new perspective on ReID system security. However, it also raises ethical and security concerns about the potential misuse of adversarial attack techniques, which could threaten public safety.
>
> Despite these concerns, adversarial attack research has positive value. It uncovers vulnerabilities in existing systems, encouraging academia and industry to improve the robustness of machine learning models. This research assesses the robustness of VIReID systems and combines adversarial training with proposed attack methods, enhancing system security and benefiting society by promoting a safer technological environment.
>
> Additionally, there is exploration of physical attacks in real-world applications. For example, AGNs[1] can create ordinary-looking glasses that cause facial recognition systems to misidentify individuals. AdvTexture[2] can cover clothing with arbitrary shapes, making individuals wearing such clothing undetectable by human detection systems. Wei et al.[3] propose using insulating materials to create physically feasible infrared patches with learnable shapes and positions, allowing pedestrians to evade infrared detectors.
>
> [1] Adversarial Generative Nets: Neural Network Attacks on State-of-the-Art Face Recognition. Arxiv 2018
>
> [2] Adversarial Texture for Fooling Person Detectors in the Physical World. CVPR 2022
>
> [3] Physically Adversarial Infrared Patches with Learnable Shapes and Locations. CVPR 2023

---

> ### Comment · Reviewer_ksiy · 2024-08-09
>
> This rebuttal addresses most of my concerns. I decide to raise my score to weak accept.

---

### Official Review · Reviewer_78wy · 2024-07-11

**Soundness:** 3
**Presentation:** 4
**Contribution:** 3
**Rating:** 6
**Confidence:** 5

**Summary:**

This paper aims to explore the security of VIReID and introduces a Universal Adversarial Perturbations to simulate common disturbances in real-world environments. Additionally, a Frequency-Spatial Attention Module is proposed to integrate frequency information extraction and spatial focusing mechanisms. An Auxiliary Quadruple Adversarial Loss is proposed to amplify the differences between modalities, thereby improving the distinction and recognition of features between visible and infrared images. Extensive experiments on two VIReID benchmarks (i.e., SYSU-MM01, RegDB) and different systems validate the effectiveness of the proposed method.

**Strengths:**

1) This paper aims to explore the security of VIReID and introduces a Universal Adversarial Perturbations to simulate common disturbances in real-world environments.

2) Extensive experiments on two VIReID benchmarks (i.e., SYSU-MM01, RegDB) and different systems validate the effectiveness of the proposed method.

3) The paper is well-written and easy to follow.

**Weaknesses:**

1) Since the author proposed a frequency domain attention module, the frequency domain and attention need to be introduced in the related work section.

2) What is the motivation of the proposed frequency domain attention module for the VIReID task?

3) The proposed frequency domain attention module uses a spatial attention module to generate spatial attention maps using the spatial relationships within the features. I'm quite curious about the performance if other attention mechanisms (such as SE-Net, Channel) are used.
Furthermore, could the authors provide the performance of the proposed frequency domain attention module on the VIReID method, thereby proving its effectiveness?

**Questions:**

Please check the weakness.

**Limitations:**

The authors have adequately addressed the limitations of the proposed method.

---

> ### Author Rebuttal · Authors · 2024-08-07
>
> **w1: The introduction of  frequency domain and attention mechanism should be added in the related work.**
>
> **Frequency domain**： In recent years, frequency domain information processing has gained significant attention in deep learning, proving effectiveness for tasks like face recognition and person re-identification. This method analyzes high and low-frequency components to extract useful information by enhancing or suppressing them. For example, PHA[1] boosts high-frequency components for better pedestrian representation. Amplitude and phase components in the frequency domain can also be used to focus on style and color versus spatial information. For instance, SFMNet[2] uses Fourier transforms for face super-resolution, capturing global image information, while Zhang et al.[3] extract modality-invariant features by leveraging amplitude and phase components.
> Attention Mechanism: Attention mechanisms allow convolutional neural networks to focus on important features while ignoring irrelevant ones. These mechanisms can be categorized into spatial, channel, and other domains. SENET[4] focuses on channel-specific features by compressing spatial dimensions and learning in the channel dimension. The Spatial Transformer Network (STN)[5] captures important regional features by transforming deformed data. CBAM[6] combines spatial and channel attention sequentially to refine image features. For visible-infrared person re-identification, we designed an attention mechanism tailored for the spatial domain.
>
>
>
> [1] PHA: Patch-wise High-frequency Augmentation for Transformer-based Person Re-identification. CVPR 2023
>
> [2] Spatial-Frequency Mutual Learning for Face Super-Resolution. CVPR 2023
>
> [3] Frequency Domain Nuances Mining for Visible-Infrared Person Re-identification VIReID. Arxiv 2024
>
> [4] Squeeze-and-Excitation Networks. CVPR 2018
>
> [5] Spatial Transformer Networks. NIPS 2015
>
> [6] Convolutional Block Attention Module. CVPR 2018
>
> **w2: The motivation of FSAM**
>
> To generate more effective adversarial features, we propose the Frequency-Spatial Attention Module based on the following:
>
> **Frequency Domain Feature Decomposition**: Visible and infrared images differ significantly in the frequency domain—visible images offer rich texture information, while infrared images contain mostly pixel data. Both share spatial consistency. We use Fast Fourier Transform (FFT) to decompose features into amplitude and phase components, capturing texture and spatial information, respectively. This approach allows targeted processing of each component, optimizing feature extraction.
>
> **Application of Spatial Attention**: Attention is applied solely to the spatial component. Since the phase component is closely tied to spatial information, the spatial attention module can enhance or suppress specific regions of the feature map. This focus improves the model's ability to highlight important areas for better adversarial feature quality and performance.
>
> **Optimization of Cross-Modal Features**:  Emphasizing spatial information helps the model integrate features from different modalities, enhancing overall comprehension and generalization.
>
> In summary, the Frequency-Spatial Attention Module improves adversarial feature quality and recognition accuracy by combining spatial attention with detailed frequency domain analysis.
>
> **w3: Replace it with channel attention**
>
> As shown in Table 2, we compared our frequency domain attention module with channel attention mechanisms like SE-Net[4] and ECA-Net[7]. The results reveal that while SE-Net and ECA-Net excel in certain areas, our frequency domain attention module is highly competitive across various metrics. It achieves the best Rank-1 and mAP scores on both datasets. Channel attention mechanisms often focus on global features, which may overlook local details and spatial relationships in cross-modal VIReID tasks, leading to potential information loss and reduced performance.
>
> |           | SYSU-MM01 |          |          |      | RegDB    |          |          |          |
> | --------- | --------- | -------- | -------- | ---- | -------- | -------- | -------- | -------- |
> | Attention | Rank-1    | Rank-10  | mAP      | mINP | Rank-1   | Rank-10  | mAP      | mINP     |
> | SENet     | 1.66      | 12.15    | 3.30     | 1.76 | 0.53     | 5.00     | 1.15     | 0.64     |
> | ECA-Net   | 1.60      | 11.65    | 3.24     | 1.65 | 0.68     | **3.64** | 1.20     | 0.68     |
> | Ours      | **0.79**  | **9.83** | **2.81** | 1.69 | **0.49** | 4.64     | **0.85** | **0.57** |
>
> [7] ECA-Net: Efficient Channel Attention for Deep Convolutional Neural Networks. CVPR 2020
>
> **w4: The effectiveness of FSAM on  VIReID**
>
> The table3 compares the performance of three VIReID systems before and after the addition of FSAM (Frequency Domain Attention Module). All systems show significant improvements in Rank-1, Rank-10, and mAP after the addition of FSAM, indicating that FSAM plays a positive role in enhancing the performance of VIReID systems. Furthermore, the consistent effects across different datasets and search modes demonstrate its broad applicability and robustness.
>
> | Methods     | SYSU-MM01 |           |           | RegDB     |           |           |
> | ----------- | --------- | --------- | --------- | --------- | --------- | --------- |
> |             | Rank-1    | Rank-10   | mAP       | Rank-1    | Rank-10   | mAP       |
> | AGW         | 47.50     | 84.39     | 47.65     | 70.05     | 86.21     | 66.37     |
> | AGW + FSAM  | **49.41** | **87.64** | **49.39** | **75.05** | **89.17** | **68.28** |
> | CAJ         | 69.88     | 95.71     | 66.89     | 85.00     | 95.50     | 84.60     |
> | CAJ + FSAM  | **70.20** | **96.36** | **67.99** | **86.77** | **95.77** | **85.01** |
> | DEEN        | 74.70     | 97.60     | 71.80     | 91.10     | 97.80     | 85.10     |
> | DEEN + FSAM | **75.20** | **97.70** | **72.27** | **92.44** | **99.09** | **86.45** |

---

> ### Comment · Reviewer_78wy · 2024-08-10
>
> Thank you for the author's response. This rebuttal addresses my concerns. I decide to change my initial score to Weak accept.

---

### Official Review · Reviewer_Jhhd · 2024-07-12

**Soundness:** 2
**Presentation:** 3
**Contribution:** 3
**Rating:** 6
**Confidence:** 3

**Summary:**

This paper addresses the security of visible-infrared person re-identification systems by introducing a method for feature-level adversarial attacks. The proposed approach integrates universal adversarial perturbations and a frequency-spatial attention module to disrupt the output ranking of VIReID systems. The auxiliary quadruple adversarial loss is designed to amplify modality differences, enhancing the distinction between visible and infrared features. Extensive experiments on the SYSU-MM01 and RegDB benchmarks validate the effectiveness of this method in compromising VIReID systems' rankings.

**Strengths:**

1. This paper is an early work in the field of the security of visible-infrared person re-identification systems.
2. The authors have made some efforts in the experimental section to validate the effectiveness of the proposed method.

**Weaknesses:**

1. The paper could benefit from a more in-depth analysis of the failure cases and limitations of the proposed method. For instance, under what conditions does the attack fail, and why?
2. The paper primarily focuses on the effectiveness of the attack in terms of ranking disruption. Additional metrics, such as computational efficiency and impact on overall system performance, could provide a more comprehensive evaluation.
3. Minor issue: Equation 15 appears to have an error.

**Questions:**

Can the authors provide more insight into the computational complexity of the proposed method?

**Limitations:**

No limitations or negative impacts have been identified in this paper.

---

> ### Author Rebuttal · Authors · 2024-08-07
>
> w1: Limitations of the proposed method**
>
> Although the proposed method is effective in many scenarios, it has certain limitations and conditions under which it may fail. These include:
>
> 1. **Extreme Imaging Conditions**: Our method relies on the alignment of adversarial features across different imaging modalities, such as visible and infrared images. However, significant variations in imaging conditions can impact the effectiveness of the feature alignment. For example, extreme lighting conditions or significant occlusions may degrade the performance of the adversarial attacks, leading to a failure in disrupting the identification process.
>
> 2. **Feature Consistency**: The proposed frequency-spatial attention module is designed to enhance the consistency of features between visible and infrared images. Nevertheless, if the inherent feature differences between the two modalities are too large, the module may not effectively bridge the gap, resulting in reduced attack efficacy.
>
> 3. **Generalizability**: Our approach has been tested primarily on specific datasets and systems, such as SYSU-MM01 and RegDB. While these datasets provide a controlled environment for evaluation, the method's performance in real-world scenarios with diverse and unseen data remains to be fully explored. Factors like variations in camera quality, resolution, and environmental conditions can influence the generalizability of the results.
>
> By acknowledging these limitations, we aim to provide a comprehensive understanding of the scope and applicability of our proposed method, thereby enhancing its utility and reliability in practical deployments.
>
> **w2 & Q: Analysis of computational complexity**
>
> According to Table 1, we analyzed the computational complexity of several methods.
>
> **Flops (Floating Point Operations)**: This metric measures the number of floating point operations required for a single forward pass of the model, reflecting its computational complexity. The Flops values of all methods in the table are quite similar, ranging from approximately 331.45G to 331.88G, indicating that these methods have comparable computational complexity.
>
> **Params (Number of Parameters)**: This metric measures the total number of trainable parameters in the model, reflecting its scale and complexity. All methods have a parameter count of 23.55M, suggesting that these methods have the same parameter scale, likely due to using the same base model or network architecture.
>
> **Conclusion**: Despite the similar computational complexity (Flops) and parameter scale (Params) across all methods, there are significant differences in their performance metrics (Rank-1, mAP, mINP). Compared to other methods, our approach has similar computational complexity and parameter count, but performs significantly lower in performance metrics. We did not drastically increase the computational complexity; instead, we achieved superior performance by balancing metrics and computational complexity more effectively.
>
> | Methods       | SYSU-MM01  |          |          |          | RegDB              |          |          | Flops       | Params |
> | ------------- | ---------- | -------- | -------- | -------- | ------------------ | -------- | -------- | ----------- | ------ |
> |               | All-search |          |          |          | Visible to Thermal |          |          |             |        |
> |               | Rank-1     | Rank-10  | mAP      | mINP     | Rank-1             | Rank-10  | mAP      |             |        |
> | Before Attack | 47.50      | 84.39    | 47.65    | 35.3     | 70.05              | 66.37    | -        | **331.45G** | 23.55M |
> | FGSM          | 36.02      | 59.22    | 31.8     | 19.72    | 45.87              | 44.39    | 36.59    | 331.75G     | 23.55M |
> | UAP           | 17.59      | 56.81    | 25.35    | 20.89    | 29.51              | 24.42    | 16.64    | 331.60G     | 23.55M |
> | Ours          | **0.79**   | **9.83** | **2.81** | **1.69** | **0.49**           | **0.85** | **0.57** | 331.88G     | 23.55M |
>
> We have carefully evaluated and optimized the computational complexity of our proposed method to ensure a balance between performance and efficiency. As indicated in Table 1, we have adopted efficient techniques to handle cross-modality data (visible and infrared), reducing the additional computational overhead typically associated with processing different data types. We utilized an optimized network architecture that balances depth and width, ensuring efficient computation without sacrificing performance.
>
> **w3: Minor issue: Equation 15 appears to have an error**
>
>  The corrected formula is:
> $$
> \[
> \mathcal{L}(x_{v2}, x_{i2}, x_{v1}) = \sum_{ \substack{j,k=1 \\ j \ne k} } \left[ D(x_{v2}^{j}, x_{v1}^{j}) - D(x_{i2}^{j}, x_{v1}^{j}) - D(x_{v2}^{j}, x_{v2}^{k}) + \alpha \right]_{+}
> \]
> $$

---

> > ### Comment · Reviewer_Jhhd · 2024-08-09
> >
> > Thanks for the author's response. I decide to raise my score to Weak Accept.

---

### Author Rebuttal · Authors · 2024-08-07

We thank all reviewers for their valuable feedback, with three reviewers (sAfc, jhhd, and 78wy) strongly supporting our work. We are pleased to see that reviewers consider our paper:

- The ideas presented are novel and interesting (Reviewer sAfc);
- The theoretical proofs are solid (Reviewer 78wy);
- Strong theoretical guarantees are provided (Reviewer jhhd);
- Extensive experiments demonstrate the effectiveness of the proposed method (Reviewer ksiy).

Reviewer ksiy's main concern is that adversarial examples may not be applicable to the real world and could cause information loss. We address this by highlighting the significance of adversarial attacks through their development history and applications across various fields, emphasizing their role in testing model robustness and providing a new perspective on system security. Additionally, we performed visualization experiments of adversarial samples and included more VIReID systems of different types to validate the effectiveness and generalizability of our method. Furthermore, we compared the effectiveness of our proposed modules through experiments on perturbed results, demonstrating the indispensability of each module.

All questions have been addressed in responses specific to each reviewer. Additionally, please refer to the attached PDF, which includes supplementary tables and figures. These are referenced and described in our individual responses to the reviewers.

---

### Decision · Program_Chairs · 2024-09-25

**Decision:**

Accept (poster)

**Comment:**

After rebuttal, all the reviewers unanimously vote for acceptance of this work. After checking the rebuttal, the review, and the manuscript, the AC recommends acceptance.